# Development of a High-Resolution Melting Method for the Detection of Clarithromycin-Resistant *Helicobacter pylori* in the Gastric Microbiome

**DOI:** 10.3390/antibiotics13100975

**Published:** 2024-10-16

**Authors:** Zupeng Kuang, Huishu Huang, Ling Chen, Yanyan Shang, Shixuan Huang, Jun Liu, Jianhui Chen, Xinqiang Xie, Moutong Chen, Lei Wu, He Gao, Hui Zhao, Ying Li, Qingping Wu

**Affiliations:** 1School of Biology and Biological Engineering, South China University of Technology, Guangzhou 510006, China; ame707059390@outlook.com (Z.K.); shangyanyan96@163.com (Y.S.); 2National Health Commission Science and Technology Innovation Platform for Nutrition and Safety of Microbial Food, Guangdong Provincial Key Laboratory of Microbial Safety and Health, State Key Laboratory of Applied Microbiology Southern China, Institute of Microbiology, Guangdong Academy of Sciences, Guangzhou 510070, China; huanghuishu0227@163.com (H.H.); chenling@gdim.cn (L.C.); nhuhuganaxis@outlook.com (S.H.); woshixinqiang@126.com (X.X.); cmtoon@hotmail.com (M.C.); wuleigdim@163.com (L.W.); gaohe.881128@163.com (H.G.); zhaohuichinese@163.com (H.Z.); 3Department of Gastroenterology, The Songgang People’s Hospital of Baoan District in Shenzhen, Shenzhen 518105, China; liusheng168@163.com; 4Division of Gastrointestinal Surgery Center, The First Affiliated Hospital of Sun Yat-sen University, Guangzhou 510080, China; slimmerchen@163.com

**Keywords:** *Helicobacter pylori*, clarithromycin, antibiotic susceptibility testing, point mutation, high-resolution melting

## Abstract

**Background:** The issue of *Helicobacter pylori* (*H. pylori*) resistance to clarithromycin (CLR) has consistently posed challenges for clinical treatment. Hence, a rapid susceptibility testing (AST) method urgently needs to be developed. **Methods:** In the present study, 35 isolates of *H. pylori* were isolated from 203 gastritis patients of the Guangzhou cohort, and the antimicrobial resistance phenotypes were associated with their genomes to analyze the relevant mutations. Based on these mutations, a rapid detection system utilizing high-resolution melting (HRM) curve analysis was designed and verified by the Shenzhen cohort, which consisted of 38 *H. pylori* strains. **Results:** Genomic analysis identified the mutation of the 2143 allele from A to G (A2143G) of *23S rRNA* as the most relevant mutation with CLR resistance (*p* < 0.01). In the HRM system, the wild-type *H. pylori* showed a melting temperature (Tm) of 79.28 ± 0.01 °C, while the mutant type exhibited a Tm of 79.96 ± 0.01 °C. These differences enabled a rapid distinction between two types of *H. pylori* (*p* < 0.01). Verification examinations showed that this system could detect target DNA as low as 0.005 ng/μL in samples without being affected by other gastric microorganisms. The method also showed a good performance in the Shenzhen validation cohort, with 81.58% accuracy, and 100% specificity. **Conclusions:** We have developed an HRM system that can accurately and quickly detect CLR resistance in *H. pylori*. This method can be directly used for the detection of gastric microbiota samples and provides a new benchmark for the simple detection of *H. pylori* resistance.

## 1. Introduction

*Helicobacter pylori* (*H. pylori*) is a prevalent pathogen in the human stomach and has been explicitly classified as a carcinogen by both the World Health Organization and the U.S. Department of Health and Human Services [1,2]. Long-term observational evidence has shown that eradication therapy for *H. pylori* infection can potentially reduce the risk of gastric cancer by 50% [3]. Since there is currently no vaccine for preventing *H. pylori* infection, antibiotics are one of the few effective treatments for *H. pylori* infection [4]. However, due to the highly acidic environment and elevated enzymatic activity in the stomach, only a limited number of antibiotics including clarithromycin (CLR), amoxicillin (AMX), levofloxacin (LVX), metronidazole (MTZ), tetracycline (TET), and rifampicin (RFP) have demonstrated efficacy in eradicating *H. pylori* in vivo [5]. Among these antibiotics, CLR is recommended as a first-line treatment in multiple global *H. pylori* treatment guidelines [6]. Nevertheless, it currently faces significant challenges associated with antimicrobial resistance (AMR), leading to an ongoing decline in success rates for eradicating *H. pylori* infection [7].

Due to the increasing global prevalence of AMR in *H. pylori*, multiple clinical guidelines recommend that physicians conduct antibiotic susceptibility testing (AST) prior to treatment for enhanced therapeutic efficacy [8,9]. However, conventional AST from a culture is time-consuming, costly, and operationally challenging, making it difficult to implement in clinical practice. In recent years, extensive research has been conducted on the AMR mechanisms of *H. pylori*, leading to the development of rapid alternative detection methods targeting resistance mechanisms as substitutes for traditional AST. Tshibangu, et al. [10] comprehensively summarized the AMR mechanisms of *H. pylori*, highlighting that gene point mutations primarily contribute to *H. pylori* resistance. Specifically, mutations occurring in the V domain of the *23S rRNA* gene serve as key genetic targets associated with CLR resistance. Furthermore, mutations in the *rpl22* and *infB* genes of *H. pylori* have also been reported to be associated with CLR resistance [11]. Thus, molecular biological detection targeting resistant genes, especially characteristic mutation sites, can partially replace traditional AST as a pre-treatment detection method for *H. pylori* eradication therapy [12].

Among the various methods for detecting mutations in the AMR genes, nucleic acid fluorescence probe technology is the most commonly employed approach. In this method, a fluorescent probe composed of adjacent fluorescent groups and quenching agents is specifically designed to bind to the target nucleic acid sequence. Upon binding to the target sequence, the fluorescent group and quenching agent separate, resulting in the release of a fluorescence signal proportional to the amount of target nucleic acid present. Subsequently, gene typing can be achieved by monitoring the dissociation process between the probe and target sequences hybridized on the amplification product during heating, and a curve graph of temperature and fluorescence signal changes can be drawn to analyze the melting curve [13]. In recent years, researchers have indicated the development of high-resolution melting (HRM) analysis based on real-time PCR (RT-PCR) for the detection of gene mutations [14]. This method utilizes saturated fluorescence dyes such as LC Green, SYTO 9, and Eva Green as substitutes for fluorescent probes. Following the completion of fluorescence quantitative PCR amplification, high-resolution melting is directly performed, and the analysis of specific gene single-nucleotide polymorphisms (SNPs) in the test samples is accomplished based on the differences in melting temperature (Tm) or the shape of the melting curve [15]. However, its applicability to *H. pylori* AST has yet to be definitively established.

In this study, a representative collection of *H. pylori* strains from the southern region of China was obtained through the establishment of a clinical cohort of gastritis patients. Whole-genome analysis was employed to identify key gene mutation sites associated with CLR resistance, and subsequently, a novel HRM detection system targeting these AMR sites was developed. This study validated the performance of the developed detection system in bacterial strain samples, gastric microbiota samples, and prospective clinical cohorts, providing a theoretical basis for the development of a simple, rapid, and cost-effective *H. pylori* AST.

## 2. Results

### 2.1. Antibiotic Resistance Profile of H. pylori

In this study, 35 strains of *H. pylori* were isolated from a cohort of 203 patients with gastric diseases, resulting in an infection rate of 17.24%. Following AMR analysis, the specific resistance profiles and MIC of the 35 *H. pylori* strains to five antibiotics were studied, as depicted in Figure 1 and Appendix A. According to the investigation, the resistance rates are as follows: 48.57% for CLR, 51.43% for LVX, 45.71% for AMX, 82.86% for MTZ, and 8.57% for TET.

### 2.2. Discovery of the CLR-Resistant Mutations

Considering that CLR is a frontline therapy for *H. pylori*, we further explored the mechanisms underlying its resistance to elucidate its molecular targets. Based on a comprehensive review by Tshibangu, we analyzed the correlation between reported mutations in the *23S rRNA*, *rpl22*, and *infB* genes and CLR resistance in 35 clinical isolates from the Guangzhou cohort (Figure 2A). Our investigation revealed that mutations conferring CLR resistance predominantly occurred at high frequencies in the *23S rRNA gene*, while mutations in the *rpl22* and *infB* genes were not observed to be associated with AMR in this cohort. Furthermore, this study found a correlation between the distribution of observed mutations in the *23S rRNA* gene and the phenotypic resistance to CLR, identifying a total of 71 mutations in this gene among the clinical isolates. Among these mutations, only the A2143G mutation at position 2143 was found to be statistically significantly associated with CLR resistance (*p* = 0.0001) (Figure 2B). WGS results revealed that among the 35 isolated *H. pylori* strains, 24 harbored the wild-type A allele at position 2143 in the *23S rRNA* gene, while 11 carried the mutant G allele. Subsequent analysis of the relationship between the A2143G mutation and the MIC of the clinical isolates demonstrated that the MIC of the mutant strain was significantly higher than that of the wild-type strain (2.67 mg/mL vs. 0.28 mg/mL, *p* < 0.0001), exceeding the clinical breakpoint for CLR resistance (Figure 2C).

### 2.3. Evaluation of HRM Detection Efficiency for the A2143G Mutation

The eight pairs of HRM primers designed in this study for detecting the *H. pylori 23S rRNA* A2143G mutation exhibited a range in Tm differences between alleles A and G at this position, from 0.27 ± 0.01 °C to 0.68 ± 0.01 °C. Among them, Primer 4 showed the highest resolution in distinguishing between the wild type and mutant types (Appendix A).

We also explored the influence of different annealing temperatures on the Tm of the two *H. pylori* genotypes in the HRM amplification reaction using Primer 4. The results revealed that within the 55–65 °C range, the Tm range for allele A was 79.34 ± 0.01 °C to 79.41 ± 0.01 °C, while for allele G, it was 79.86 ± 0.01 °C to 79.93 ± 0.1 °C, indicating that the annealing temperature within this range did not affect Tm interpretation (*p* > 0.05). Visual differentiation of the standardized melting peaks after conversion to a different plot allowed for the straightforward discrimination of the two genotypes (Appendix A).

Subsequent validation of the accuracy of HRM detection using Primer 4 revealed that among the 35 isolated *H. pylori* strains, 24 exhibited a Tm of 79.28 ± 0.01 °C, predicting allele A, while 11 strains exhibited a Tm of 79.96 ± 0.01 °C, predicting allele G (Figure 3A,B). Comparison with whole-genome sequencing results confirmed a 100% match between the predictions of alleles A and G in the *23S rRNA* gene at position 2143 using the Primer 4 HRM system.

Statistical analysis indicated that the Tm of the 24 strains with the *23S rRNA* gene allele A in the HRM reaction system was 79.28 ± 0.01 °C, while that of the 11 strains with the *23S rRNA* gene allele G was 79.96 ± 0.01 °C. The difference in Tm between different gene alleles within the range of 0.68 ± 0.01 °C was statistically significant (*p* < 0.0001) (Figure 3C).

Detection of samples at different concentrations revealed that for the *23S rRNA* gene allele A strains, template DNA formed a melting peak at 79.35 ± 0.01 °C within the concentration range of 50 ng/μL to 0.005 ng/μL, while no melting peak Tm was observed when the concentration decreased to 0.0005 ng/μL. Similarly, for the *23S rRNA* gene allele G strains, template DNA formed a melting peak at 79.96 ± 0.01 °C within the concentration range of 50 ng/μL to 0.005 ng/μL, but no melting peak Tm was observed when the concentration decreased to 0.0005 ng/μL (Figure 3D). This result suggests that the detection limit of this HRM detection system is 0.005 ng/μL.

### 2.4. Detection Performance of the HRM System in the Gastric Microbiome

This study further validated the detection efficiency of the HRM detection system in gastric microbiota samples. Here, 14 gastric microorganisms (except *H. pylori*) belonging to 5 genera and 14 species were isolated from gastric mucosal samples (Appendix A). Utilizing the reaction system based on Primer 4, the DNA of these strains was subjected to HRM detection, revealing the following Tm values for the 14 strains: 80.84 ± 0.01 °C (*Streptococcus anginosus*), 82.56 ± 0.01 °C (*Streptococcus parasanguinis*), 80.83 ± 0.01 °C (*Streptococcus salivarius*), 81.21 ± 0.01 °C (*Streptococcus pseudopneumoniae*), 80.45 ± 0.01 °C (*Streptococcus oralis*), 81.27 ± 0.01 °C (*Streptococcus sanguinis*), 81.06 ± 0.03 °C (*Bacillus nealsonii*), 80.75 ± 0.01 °C (*Bacillus aryabhattai*), 82.25 ± 0.01 °C (*Bacillus stratosphericus*), 81.07 ± 0.03 °C (*Bacillus altitudinis*), 81.52 ± 0.01 °C (*Rothia dentocariosa*), 81.24 ± 0.01 °C (*Rothia mucilaginosa*), 81.39 ± 0.01 °C (*Staphylococcus warneri*), and 80.81 ± 0.01 °C (*Gemella sanguinis*). These Tm values exhibited significant differences when compared with the Tm values of *H. pylori* strains with the A or G allele at position 2143 (*p* < 0.05) (Figure 4A,B).

Furthermore, detection of simulated gastric microbiota samples revealed that in samples containing 10%, 30%, 50%, 70%, and 90% A allele of *H. pylori*, the Tm values were 79.20 ± 0.01 °C, 79.26 ± 0.01 °C, 79.26 ± 0.01 °C, 79.26 ± 0.01 °C, and 79.20 ± 0.01 °C, respectively, showing no statistically significant differences compared to pure bacterial samples (*p* < 0.05). In samples containing 10%, 30%, 50%, 70%, and 90% G allele of *H. pylori*, the Tm values were 79.98 ± 0.01 °C, 79.98 ± 0.01 °C, 79.98 ± 0.01 °C, 79.98 ± 0.01 °C, and 79.98 ± 0.01 °C, also exhibiting no statistically significant differences compared to pure bacterial samples (*p* > 0.05) (Figure 4C).

Moreover, HRM analysis of mixed samples of two genotypes showed distinct standardized melting curves for the wild-type, mutant, and mixed samples. Unlike the single peaks observed in pure samples, the mixed samples formed two melting peaks at 78.31 °C and 79.99 °C, indicating the presence of two mixed microbial genotypes. This suggests that the method is capable of identifying and detecting clinical samples with heteroresistance (HR) (Figure 4D).

### 2.5. Detection Efficiency of the HRM System in the Prospective Clinical Cohort

To assess the viability of the HRM system in clinical diagnosis, we conducted further validation by applying it to clinical samples obtained from a cohort of 38 cases in Shenzhen. In this study, we compared the consistency of results obtained through phenotypic experiments, genomic sequencing, and HRM analysis within this cohort (Figure 5A). The determination agreement between HRM and sequencing methods for the 2143 allele of *H. pylori 23S rRNA* gene was found to be 100%. However, when comparing HRM with phenotypic experiments, the concordance rate for determining CLR resistance was only 81.58%. Our comparison of MIC values for strains with Allele A and Allele G at position 2143 revealed a statistically significant difference (*p* < 0.0001), with a mean MIC value of 1.06 ± 0.29 mg/L for Allele A strains and 3.40 ± 0.29 mg/L for Allele G strains (Figure 5B). In this validation cohort, the sensitivity was determined to be 62.50%, while the specificity was found to be 100.00%. By constructing an ROC curve for HRM detection, we obtained an AUC value of 0.59 (Figure 5C). These findings suggest that our method exhibits excellent specificity and demonstrates a high accuracy in predicting negative samples.

## 3. Discussion

*H. pylori* is a prevalent and potentially carcinogenic microorganism globally, necessitating eradication therapy with antibiotics following infection [16,17]. However, the rampant misuse of antibiotics in the environment, food, and medical sectors has precipitated rapid development of resistance in *H. pylori*, resulting in a sharp decline in the efficacy of anti-infective treatments [12]. Epidemiological studies have revealed that the resistance rates of *H. pylori* have been in a state of constant flux across different regions and years. In this study, we assessed the AMR prevalence of *H. pylori* in Southern China during 2021. Our findings indicated a dynamic state in the resistance rates of CLR (48.57% vs. 50.00%, *p* = 0.510), LVX (51.43% vs. 33.33%, *p* = 0.089), MTZ (82.86% vs. 77.78%, *p* = 0.195), AMX (45.71% vs. 18.52%, *p* = 0.161), and TET (8.57% vs. 12.96%, *p* = 0.104) compared to the rates in 2020 in this region [18]. Consequently, it is imperative to develop a simple AST method for monitoring the dynamic change in AMR in order to appropriately select antibiotics [19].

The traditional method for detecting AMR of *H. pylori* is challenging, time-consuming, and costly, making it difficult to be widely implemented in clinical practice. In recent years, the rapid advancement in the field of nucleic acid testing has made genotype-based AST the mainstream direction for resistance detection [20,21]. The identification of appropriate molecular targets is crucial for achieving precise genotype-based AST. Unlike many common microorganisms, where AMR phenotypes are mediated by explicit resistance genes, gene mutations primarily drive the molecular mechanisms of *H. pylori* resistance [22]. Through genomic analysis of 35 cases of *H. pylori* in the Guangzhou cohort, we discovered a significant correlation (*p* = 0.001) between the A2143G mutation in the *23S rRNA* gene and CLR resistance. This site is a critical mutation point in the V structure domain of the *23S rRNA* gene, and its mutation diminishes the binding affinity of CLR to the bacterial strain, rendering it ineffective. This mutation has also been frequently mentioned in epidemiological studies in other regions. Further comparison revealed that samples with the A2143G mutation generally exhibited higher MIC values than the wild-type samples. Consequently, we selected this mutation as the target for subsequent genotype-based testing. It is noteworthy that, contrary to other literature reports, our analysis did not find a correlation between the *23S rRNA* A2142G/C mutation and resistance, suggesting that personalized molecular biology testing may be necessary based on the results of previous epidemiological investigations in different regions.

The development of detection methods should move towards low cost, high specificity and sensitivity, and being applicable to the complex gastric microbiota. Therefore, we have developed an AST detection system for *H. pylori* based on the HRM method. Previous studies have indicated that the choice of intercalating dye is a crucial factor influencing HRM detection. Compared to the unsaturated dye SYBR Green, intercalating dyes can achieve a higher reaction efficiency [23,24]. Therefore, this study employed the saturated dye Evagreen. Additionally, the length of the amplicon is a major factor determining the resolution and specificity of HRM analysis: a shorter PCR product can improve HRM resolution, but excessively short products can lead to non-specific sequence amplification. On the other hand, excessively long PCR products can make it difficult to distinguish individual base mutations [25]. Thus, in designing HRM primers, in addition to following the general principles of RT-PCR primer design, it is necessary to consider the length of the amplified product, while positioning SNP sites as close to the middle of the PCR product sequence as possible, ensuring both amplification efficiency and the elimination of interference from non-specific sequence amplification. The annealing temperature directly affects the specificity and sensitivity of the amplification reaction. A lower annealing temperature may increase the sensitivity of PCR by reducing the chance of non-specific binding between primer and template, thereby enhancing the strength of the detection signal. However, an excessively low annealing temperature can decrease the specificity of the reaction, as the primer may have a strong binding affinity with regions it should not bind to (e.g., non-target sequences) [26,27,28]. Following a series of explorations in primer design and annealing temperature for the amplification reaction, we ultimately achieved a resolution of 0.68 ± 0.01 °C for the melting peak temperature in HRM, and this difference in melting was optimized by the difference plot algorithm to form a more intuitive curve shape. Tm difference is pivotal in influencing the outcomes of an HRM-based assay. Research indicates that the majority of class I SNPs (A to G or C to T mutations) typically result in a significant Tm difference exceeding 0.5 °C [29,30]. Melting devices with high thermal accuracy excel at differentiating between bases, and yet we might need to contemplate enhancing the precision by certain means to adapt it for a wider range of ordinary laboratories. Currently, some researchers have proposed improvement strategies such as competitive amplification of differently melting amplicons to ensure a minimum difference of 1 °C in Tm. These methods could be used to genotype SNPs effectively in any qPCR devices including low-thermal-precision instruments [29,31]. The detection scheme we developed was validated in 35 *H. pylori* strains from the Guangzhou area, demonstrating that HRM can achieve a high accuracy of 100% in distinguishing A/G bases, and the detection sensitivity can reach as low as 0.005 ng/μL. More importantly, through validation with artificially contaminated gastric microbiota samples, our developed HRM detection system effectively identified wild-type and mutant *H. pylori* directly in these samples, indicating its applicability for culture-free resistance testing and greatly simplifying the current process of *H. pylori* AST [32]. Additionally, we have included the complete HRM testing process in Appendix A.

Subsequent prospective clinical cohort studies, however, revealed limitations in the efficacy of this approach. While the HRM-based AST detection method in this cohort was able to accurately identify the A/G genotype at position 2143 of the *23S rRNA* gene with 100% specificity, its sensitivity was only 62.50% for predicting CLR resistance not caused by the A2143G mutation. We hypothesize that in actual samples, *H. pylori* exhibits various mechanisms for CLR tolerance, including mutations at other genetic loci, drug efflux pumps, or biofilm formation, in addition to the A2143G mutation in the *23S rRNA* gene. The current single-target detection system inevitably leads to this phenomenon of missed detection, which is responsible for the suboptimal ROC curves observed in our validation cohort. To address this challenge, the exploration of novel resistance targets and the development of multi-target detection systems are the directions we need to consider [33].

## 4. Materials and Methods

### 4.1. Study Cohorts and Sample Collection

A cohort of 203 gastritis patients were enrolled in the First Affiliated Hospital of Sun Yat-sen University in Guangzhou, Guangdong, China from 1 June to 31 October 2021. Another cohort of 292 gastritis patients were enrolled in the Songgang People’s Hospital of Baoan District in Shenzhen, Guangdong, China from 1 June to 1 August 2022. All gastric biopsy samples were collected during the endoscopic examination according to the protocol of Yamaoka [34]. Written informed consent was obtained from all patients, and this study was conducted according to the guidelines of the Declaration of Helsinki and approved by the ethics committees of the First Affiliated Hospital of Sun Yat-sen University (project number 2020-164) and the Songgang People’s Hospital of Baoan District in Shenzhen (project number 2022-406).

### 4.2. Isolation and Identification of the Gastric Microbes

The gastric mucosa was homogenized, and the suspension was inoculated on a Columbia-based blood agar plate (Huankai Microbial, Guangzhou, China) at 37 °C in a microaerophilic incubator (Binder, Tuttlingen, Germany) containing 10% O_2_, 5% CO_2_, and 85% N_2_ for 7~10 days. The microbial colonies growing on the plates were picked and transferred to another Columbia-based blood agar plate for purification in the microaerophilic incubator for another 72 h. The purified colonies were washed and resuspended in phosphate-buffered saline for DNA extraction and AST.

Genomic DNA was extracted from the gastric microbes using a genomic DNA extraction kit (Huankai) and their taxonomic analysis was carried out by Sanger sequencing using the universal primers of the *16S rRNA* gene (27F: 5′-AGAGTTTGATCCTGGCTCAG-3′, 1492R: 5′-ACGGCTACCTTGTTACGACTT-3′) [35]. The sequences were compared with reference sequences from GenBank using the BLAST tool (http://www.ncbi.nlm.nih.gov/BLAST (accessed on 13 February 2023)) for taxonomic identification.

### 4.3. Phenotypic AST of H. Pylori

The antibiotic susceptibility phenotype was identified using the agar dilution assay according to the guidelines of the Clinical and Laboratory Standards Institute [36]. Five antibiotics were used in this study: AMX, CLR, LVX, MTZ, and TET (Meilunbio, Dalian, China), and *H. pylori* ATCC 43504 was used as the quality control. The breakpoints of minimum inhibitory concentration (MIC) were set up for discriminating susceptible and resistant phenotypes according to the European Committee on Antimicrobial Susceptibility Testing [37], in which MICs exceeding 0.125 mg/L for AMX, 0.25 mg/L for CLR, 1 mg/L for LVX, 8 mg/L for MTZ, and 1 mg/L for TET indicated AMR of *H. pylori*.

### 4.4. Sequencing

#### 4.4.1. Whole-Genome Sequencing (WGS) and Genome Assembly of *H. pylori*

Genomic DNA libraries were generated using the AMT Rapid DNA-Seq Kits from Illumina (Cistro, Guangzhou, China), with fragmentation, end-repair, adaptor ligation, size selection, and amplification according to the manufacturer’s instructions. The libraries were assessed using a Bioanalyzer 2100 (Agilent, Santa Clara, CA, USA) and a Qubit 3.0 fluorometer (Invitrogen, Waltham, MA, USA). DNA sequencing was performed on the Nextseq 550 platform (Illumina, San Diego, CA, USA) with a High Output v2.5 kit (Illumina). Low-quality reads from Illumina sequencing were first filtered using Trimmomatic (v0.39) [38], and then they were aligned into de novo assembled contigs using SPAdes (v0.4.8) [39].

#### 4.4.2. Gastric Microbiome Analysis

Gastric microbiome analysis was carried according to the method of Li [18]. In brief, microbial DNA of the gastric mucosa was extracted using the QIAamp PowerFecal Pro DNA kit (Qiagen, Hilden, Germany). *16S rRNA* gene amplicon libraries were built using the V3–V4 hypervariable region primers 338F 5′-ACTCCTACGGGAGGCAGCAG-3′ and 806R 5′-GGACTACHVGGGTWTCTAAT-3′ [40]. Paired-end sequencing was conducted on the MiSeq platform (Illumina) with the MiSeq Reagent Kit V3 (Illumina). The raw data were trimmed and quality-filtered using the CLC genomic workbench 20.0 (Qiagen) and compared to the reference sequences in SILVA 138.1 for taxonomic analysis.

### 4.5. Bioinformatic Analysis of the AMR Genetic Determinants

Bioinformatic analysis of the AMR genetic determinants was performed according to the method of Li [18]. In brief, the genomes of the *H. pylori* strains were annotated using Prokka v1.11 [41]. A core SNP alignment tree was generated using Parsnp [42]. The AMR genotypes were assessed using the CLC genomic workbench v20.0 (Qiagen): targeted genes including *23S rRNA*, *rpl22*, and *infB* were retrieved from the complete genome of *ATCC 26695* as references, and variant identification was carried out among the isolates and the reference using the CLC genomic variant detection module.

### 4.6. Primer Design and HRM Analysis

Primer Premier v6.0 (Premier Biosoft, Palo Alto, CA, USA) was used to design 8 pairs of primers for the *23S rRNA* gene amplicon ranging from 50 bp to 121 bp to identify the AMR alleles (Appendix A), and the resolution and specificity of these primers were compared by their melting peak and difference plot.

The HRM analysis for the mutation examination was carried out using the HRM analysis kit (Tiangen Biotechnology, Beijing, China) on the LightCycler 96 Instrument (Roche, Branchburg, NJ, USA). The total volume of the HRM reaction system was set to 20 μL, containing 10 μL 2× HRM analysis premix, 1 μL 1pmol/μL for each primer, and 1 μL 50 ng/μL DNA template. Five annealing reaction temperatures were set in the range of 55~65 °C to explore the best reaction conditions of HRM amplification. The procedures of HRM were as follows: pre-denaturation at 95 °C for 2 min; PCR reaction at 95 °C for 10 s, 55/57/60/63/65 °C for 20 s, and 72 °C for 30 s, in alternating cycles 40 times. After amplification, high-resolution melting curve procedures were added: 95 °C for 60 s, 40 °C for 60 s, 55 °C~65 °C for 1 s, and 97 °C for 1 s, and the primary fluorescence was collected at 0.02 °C~0.10 °C during the melting curve analysis. LightCycler 96 gene scanning software v.1.1 (Roche) was used for signal normalization, temperature shift, and difference plot analysis.

### 4.7. Verification of the Detection Efficiency of the HRM System

The detection accuracy, sensitivity, and specificity were systemically verified in this study. The accuracy of the HRM system was examined in the 35 *H. pylori* isolates from the Guangzhou cohort by comparing the predicted results of the system to the known allele of WGS. DNA samples of both the AMR strain and the CLR-susceptive strains were diluted into six concentrations for sensitivity verification of the HRM system, including 50 ng/μL, 5 ng/μL, 0.5 ng/μL, 0.05 ng/μL, 0.005 ng/μL, and 0.0005 ng/μL. The specificity of the HRM system was examined in several gastric microbes isolated from gastric mucosal samples, 10 gastric microbiome samples that were artificially contaminated by different concentrations (10%, 30%, 50%, 70%, 90%) of CLR-susceptible and -resistant *H. pylori* DNA, as well as a combined sample of CLR-susceptible and -resistant (DNA concentration 1:1) *H. pylori*. Finally, the detective efficiency of the HRM system was verified in the 38 *H. pylori* isolates from the Shenzhen cohort by comparing the predicted results of the HRM system to the AMR phenotypes of the strains.

### 4.8. Statistical Analysis

Data were analyzed using PASW Statistics 18.0.0 (IBM, NY, USA). A comparison of the Tm in different samples was carried out using *T*-test, while MICs in different AMR phenotypes were compared using the Mann–Whitney U-test. The association between the genetic mutations and AMR phenotypes were analyzed using the χ^2^ test. *p* < 0.05 was considered statistically significant.

## 5. Conclusions

In conclusion, our work demonstrates that single-base typing based on HRM is a method suitable for clinical detection of CLR resistance in *H. pylori*. We believe that after further expansion of the validated case numbers, this method will exhibit a greater detection performance. Additionally, the integration of this work with different clinical cohorts has proven to be an indispensable part of advancing research in the field of *H. pylori*. Combining retrospective studies with prospective validations will further broaden the applicability of the detection system and enhance the interpretation of the results, holding significant implications for broader adoption.

## Figures and Tables

**Figure 1 antibiotics-13-00975-f001:**
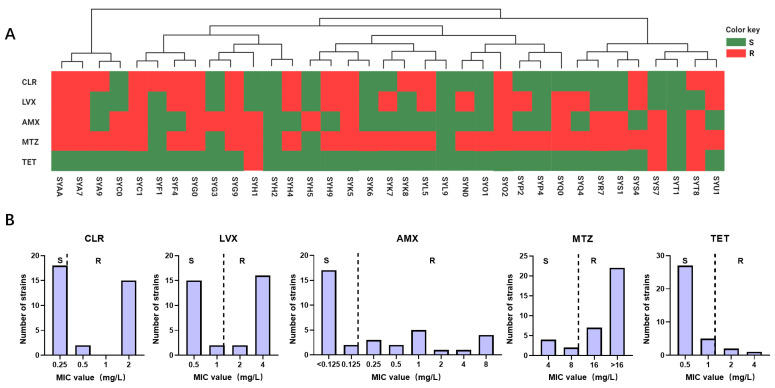
Epidemiological investigation of *H. pylori* antibiotic resistance in the Guangzhou cohort. (**A**) The resistance profiles of 35 isolated *H. pylori* strains to five antibiotics are depicted. Phylogenetic trees were constructed based on SNP analysis of the genomic sequences, with each *H. pylori* strain represented by rectangular bars in green (susceptible) and red (resistant), corresponding to their antibiotic resistance phenotypes. (**B**) MIC values of *H. pylori* isolated strains from the Guangzhou cohort for 5 antibiotics are presented, denoted as R and S.

**Figure 2 antibiotics-13-00975-f002:**
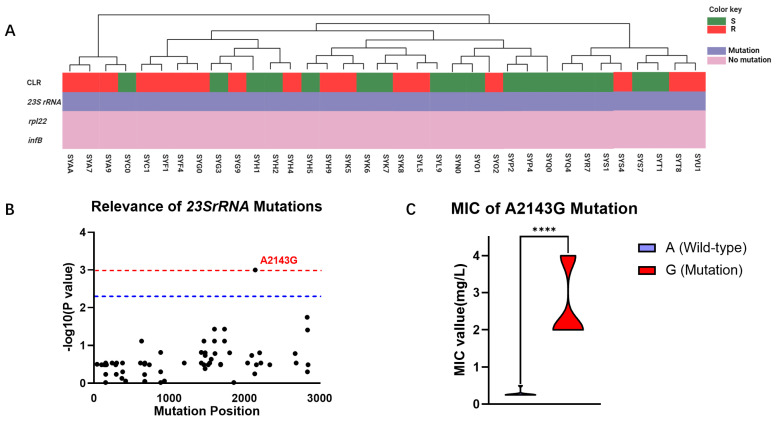
Mutational profiles of 35 clinical isolates from the Guangzhou cohort. (**A**) Genetic analysis of CLR resistance in the Guangzhou cohort: The phenotype of strain resistance to CLR is represented by green (S) and red (R) rectangles. Purple indicates the presence of gene mutations, while pink indicates the absence of mutations. (**B**) Correlation analysis of different mutational sites in the *23S rRNA* gene of the Guangzhou cohort with CLR resistance phenotypes: The horizontal axis repre-sents the positions of mutational sites in the *23S rRNA* gene, while the vertical axis indicates the likelihood of their association with CLR resistance phenotypes, with *p*-values represented as −log10. The blue line denotes significant correlation (*p* = 0.005), and the red line denotes highly significant correlation (*p* = 0.001). (**C**) MIC comparison of *H. pylori* with A and G alleles at position 2143 in the *23S rRNA* gene: Purple represents samples with the A2143G mutation, while red represents wild-type samples without the A2143G mutation. ****: *p* < 0.0001.

**Figure 3 antibiotics-13-00975-f003:**
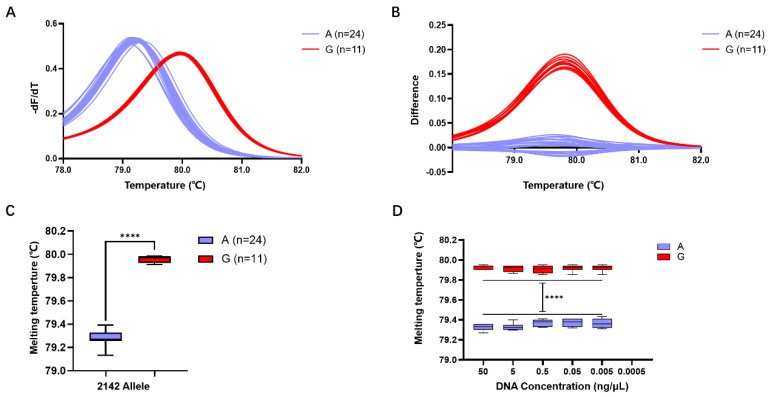
Evaluation of *H. pylori 23S rRNA* gene A2143G detection efficiency using the HRM system based on Primer 4. (**A**) Standardized melting peaks detected in 35 isolated *H. pylori* strains by the HRM system. (**B**) The optimized difference plot formed from the standardized melting peaks after algorithm optimization by the LightCycler^®^ 480 PCR instrument, allowing for more intuitive differ-entiation. (**C**) Statistical analysis of the Tm values for samples with the 2143 allele A (purple) and the 2143 allele G. (**D**) Tm values of genomic DNA from samples with 2143 allele A (purple) and 2143 allele G (red) at different concentrations. ****: *p* < 0.0001.

**Figure 4 antibiotics-13-00975-f004:**
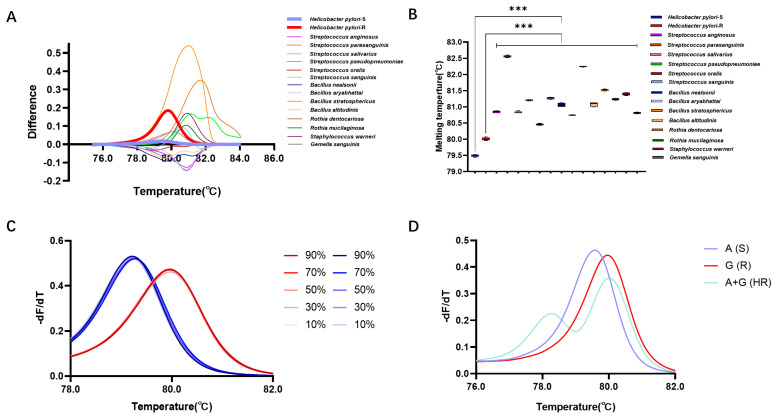
Detection performance of HRM in gastric microbiota. (**A**) Difference plot curves of differ-ent gastric isolated strains. (**B**) Tm of different gastric isolated strains. (**C**) Normalized melting peaks at different abundances of *H. pylori*. (**D**) Standardized melting peaks of wild-type, resistant-type, and two mixed microbial populations. ***: *p* < 0.001.

**Figure 5 antibiotics-13-00975-f005:**
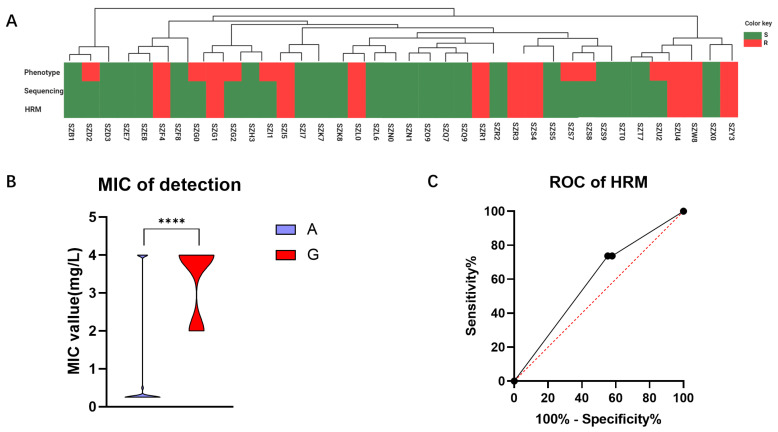
Validation outcomes of the Shenzhen queue samples. (**A**) To validate the detection results of the queue, a comparative analysis was conducted among the phenotype experiment, sequencing, and HRM methods. S is represented by the green color, and R by the red color. (**B**) The relationship between sensitive and resistant samples in the validation cohort with MIC is depicted. (**C**) An ROC curve was constructed based on HRM detection results obtained from the validation queue. ****: *p* < 0.0001.

## Data Availability

The extracted data that support the findings of this study are available from the Guangdong Academy of Sciences Institute of Microbiology (GDIM) database, but restrictions apply to the availability of these data, which were used under a license for the current study and thus are not publicly available.

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
