# Peer review of "Development of a High-Resolution Melting Method for the Detection of Clarithromycin-Resistant Helicobacter pylori in the Gastric Microbiome"

_antibiotics, 2024, doi:10.3390/antibiotics13100975_

Round 1
Reviewer 1 Report
Comments and Suggestions for Authors
I reviewed a narrative review article titled: "Development of a high resolution melting method for the detection of clarithromycin resistant Helicobacter pylori in gastric microbiome". Minor revision seems to be required for publication.
In Figure 1(A), the colors of red, tan, and green were mismatched with antibiotic resistance status including resistant (R), intermediate (I), and susceptible (S).
Please confirm whether MIC values of five antibiotics are consistent with those shown in Figure 1(B).
Reviewer 2 Report
Comments and Suggestions for Authors
After reading the content of the original article entitled “Development of a high resolution melting method for the detection of Clarithromycin resistant Helicobacter pylori in gastric microbiome”, I believe that the article is very interesting and the subject undertaken is important. There are few issues that should be addressed. The list is provided below:
- Title: “Clarithromycin” -> should be written in lower case
- It is possible that I misunderstood something, but in some places of the article the authors mention that they tested 35 strains of H. pylori and in others that they tested 38 stains (please clarify and correct any places in the manuscript if necessary)
- Please correct the citation style in the text according to the MDPI guidelines
- Line 31: “100%specificity” -> 100% specificity
- The authors provided an incorrect list of references, which does not correspond to the equivalents quoted in the text, e.g. refs 32 and 33. Please correct it without fail.
- When classifying antibiotic resistance, the authors refer to EUCAST, but these recommendations certainly do not include the "intermediate" classification, which the authors introduced in the case of some antibiotics - please provide explanation and possible correction in all appropriate places in the manuscript
Reviewer 3 Report
Comments and Suggestions for Authors
In this manuscript, authors identified the mutation in 23S rRNA of H. pylori that is associated with CLR resistance, and developed HRM system to identify it. Authors performed many necessary experiments and presented the results appropriately. This paper will provide practically important information to readers and contribute to develop methods to detect CLR resistance in HP. Manuscript is almost very well written with necessary results in Figures. This reviewer suggests following points to improve the manuscript.
1. Figure 1A, 2A, 4A/B, 5A: letters in vertical and horizontal lines are too small to read. Enlarge somewhat.
2. In Fig.1A, green square is indicated as "R", in the right side example. Is it correct?
3. Figure 2, 3: Legend s described as "Depicts ....". This is uncommon descriptions. Please use more common wordings.
4. Legend of Figure 3(D) (line 182-183): shows allele G (purple)....allele A (red). However, in Figure 4 (D), these are reversed Please correct.
5. Throughout the manuscript, please confirm italic form for "H. pylori". Some portions are not italicized. CLR is sometimes shown as clarithromycin. Please keep uniformity.
6. Authors identified melting temperature to discriminate A2143G. It is wonderful finding. However, the difference is very minimal. In such case, the laboratory/qPCR machine with highly sophisticated accuracy of the temperature appears to develop the one-nucleotide difference. In contrast, it seems that the HRM method is not available in ordinary laboratory. This reviewer hopes to see any opinion or idea in this regard in Discussion.
7. This manuscript is highly contributable to clinical doctors and laboratory, to detect A2143G mutation by the HRM method. In such case, reproducibility of this method would be very important, and will add reliability of this paper. However, it seems that the protocol of HRM, which author concluded as the best scheme, is not clear. Probably, readers may follow the description in 2.3 section, using primer 4, and also referring to Method section. Though these may be enough, it is preferable to add the method/protocol for readers to attempt the HRM method as supplementary material, to confirm reproducibility and application to clinical specimens.
Round 2
Reviewer 2 Report
Comments and Suggestions for Authors
I would like to thank Authors for following all the suggestions and applying them in the manuscript. I believe that its quality has improved.